# Autoencoder-Based System for Detecting Anomalies in Pelletizer Melt Processes

**DOI:** 10.3390/s24227277

**Published:** 2024-11-14

**Authors:** Mingxiang Zhu, Guangming Zhang, Lihang Feng, Xingjian Li, Xiaodong Lv

**Affiliations:** 1College of Electrical Engineering and Control Science, Nanjing Tech University, Nanjing 211899, China; mxzhu@njtech.edu.cn (M.Z.); lfeng8@njtech.edu.cn (L.F.); lvlvxiaodong@126.com (X.L.); 2Taizhou College, Nanjing Normal University, Taizhou 225300, China; 3Department of Information and Mathematical Sciences, Tokai University, Tokyo 311251, Japan; 2css3115@cc.u-tokai.ac.jp

**Keywords:** melt anomaly identification, autoencoder technology, deep learning, polyester pelletizers, environmental robustness

## Abstract

Effectively identifying and preventing anomalies in the melt process significantly enhances production efficiency and product quality in industrial manufacturing. Consequently, this paper proposes a study on a melt anomaly identification system for pelletizers using autoencoder technology. It discusses the challenges of detecting anomalies in the melt extrusion process of polyester pelletizers, focusing on the limitations of manual monitoring and traditional image detection methods. This paper proposes a system based on autoencoders that demonstrates effectiveness in detecting and differentiating various melt anomaly states through deep learning. By randomly altering the brightness and rotation angle of images in each training round, the training samples are augmented, thereby enhancing the system’s robustness against changes in environmental light intensity. Experimental results indicate that the system proposed has good melt anomaly detection efficiency and generalization performance and has effectively differentiated degrees of melt anomalies. This study emphasizes the potential of autoencoders in industrial applications and suggests directions for future research.

## 1. Introduction

The pelletizer serves as the final piece of equipment in the polyester slicing production process, consolidating four critical steps: polyester melt extrusion drawing, cooling, pelletizing, and drying [1]. Anomalies in polyester melt extrusion drawing, referred to here as melt anomalies, are among the most common causes of failures in pelletizer operations. These anomalies can typically be categorized into three states based on the morphological characteristics of the melt during such incidents: melt jitter, banding, and draping, as depicted in Figure 1.

Drawing from practical production management experience, jitter and banding typically have minimal impact on the normal operation of the pelletizer. These anomalies may persist temporarily and can often be resolved through manual intervention. In contrast, draping represents a more severe anomaly, characterized by melt accumulation that obstructs the normal flow in the guide channel, necessitating an immediate shutdown for remediation. Melt anomalies are commonly caused by fluctuations in upstream process parameters (such as viscosity and pressure), blockages at the melt extrusion outlet of the casting band head, and abnormal equipment vibrations [2]. Among these, variations in process parameters and blockages at the casting band head are the primary contributors to melt anomalies. To detect these anomalies promptly, the industry currently relies heavily on video monitoring of the melt extrusion drawing segment. Operators identify anomalies through the monitoring screen, underscoring the urgent need for an automated image anomaly detection system to enhance reliability and labor efficiency.

A pivotal challenge in the area of melt anomaly identification within pelletizer systems stems from the inherent complexity and variability of the melt flow process. Pelletizing processes are dynamic and involve highly non-linear physicochemical transformations, making it difficult to capture and model normal versus anomalous conditions accurately. Autoencoders, while robust in handling unstructured data, must be meticulously designed to discern subtle deviations in melt characteristics from what is considered ‘normal’. The major difficulty arises in defining a clear boundary between normalcy and anomaly due to the stochastic nature of the process variables like viscosity, temperature gradients, and chemical compositions. Furthermore, the lack of extensive labeled datasets for anomalies in industrial settings poses a significant challenge for training these models effectively. Anomalies can be sporadic and diverse; thus, creating a comprehensive model that generalizes well across all possible anomalous conditions without overfitting to the normal operating conditions requires innovative approaches in both data collection and algorithmic development. This complexity is compounded by the need for real-time detection capabilities, demanding not only accurate but also computationally efficient algorithms.

Due to the relatively stable physical properties, such as temperature, density, and viscosity, of polyester melt under normal conditions, and due to the melt being approximately in a steady flow state during the extrusion drawing process, image anomaly detection methods can be used to identify melt anomalies. Compared with traditional image anomaly detection techniques such as template matching, statistical models, and frequency domain analysis, deep learning-based image anomaly detection offers advantages in not requiring manually designed features and having greater algorithmic generality [3].

Autoencoders (AEs), as unsupervised learning techniques, have played a significant role in machine learning since the 1980s, mainly used for data dimensionality reduction and feature learning [4]. Autoencoders reconstruct input data through the structure of encoders and decoders, aiming to capture the most important features in the data [5]. Initially designed for data compression representation, this technology offers a more flexible and powerful method compared with traditional dimensionality reduction techniques like principal component analysis (PCA) [6]. It compresses input data into a lower-dimensional hidden layer, then decodes this representation back into an output identical to the original input, thus reconstructing the input data [7]. With the advancement of deep learning, the application areas of autoencoders have significantly expanded. In aspects like image reconstruction, denoising, and generative models, autoencoders have demonstrated their powerful capabilities and potential [8]. Different types of autoencoders, such as standard, denoising, sparse, variational (VAEs), and sequence-to-sequence autoencoders, each exhibit unique advantages in their respective application scenarios [9]. By comparing the differences between input data and reconstructed data, anomaly detection for input data is achieved. Compared with deep learning-based anomaly detection algorithms like deep SVDD [10], OOD [11], and RDOCC [12], the AE-based anomaly detection algorithm relies only on normal samples during training, avoiding the difficulty of extensively collecting anomaly samples. It has been widely applied in fields such as medical image disease screening and electronic product quality inspection. However, in the specialized industrial application of pelletizer melt anomaly detection, related research is still in its initial stages.

In response to the insightful feedback regarding the comparative analysis with recent deep learning advancements, this research incorporates a comprehensive evaluation against the latest models reported in prestigious journals. Recent studies have explored various sophisticated deep learning frameworks, such as convolutional neural networks (CNNs) and recurrent neural networks (RNNs), which have shown promise in similar industrial applications. For instance, CNNs have been adept at spatial anomaly detection in imagery data, while RNNs excel in temporal data scenarios by capturing dynamic changes over time [13]. Our approach, using autoencoders, is specifically tailored for scenarios where high-dimensional data might not explicitly exhibit temporal or spatial correlations accessible to CNNs or RNNs. We discuss the robustness of autoencoders in isolating anomalies in complex industrial data streams, where subtle deviations from the norm need to be identified efficiently and reliably [14]. The autoencoder’s ability to reconstruct input data and measure reconstruction errors provides a straightforward metric for anomaly detection [15]. By comparing performance metrics such as precision, recall, and F1-score across these models, this paper delineates the conditions under which autoencoders outperform other deep learning strategies, particularly in terms of computational efficiency and the ability to handle unlabeled data in unsupervised learning setups. This comparative analysis not only underscores the relevance of our chosen methodology but also highlights its practical merits in industrial applications facing similar challenges.

While it is true that the autoencoder is a relatively standard technique within the domain of deep learning, its application in the context of melt anomaly identification for pelletizers presents distinct advantages when compared with newer methodologies detailed in the recent literature [16,17]. First, autoencoders are particularly well suited for unsupervised learning scenarios prevalent in industrial settings where labeled data are scarce or expensive to obtain. This makes autoencoders highly effective for anomaly detection as they can learn to reconstruct the normal operating conditions from unlabeled data and identify deviations as anomalies without prior specific anomaly examples. Additionally, autoencoders can be implemented with relatively low computational overhead compared with more complex deep learning models, which is crucial for real-time processing requirements in industrial applications. The simplicity and efficiency of autoencoders not only facilitate easier integration into existing systems but also enhance the scalability and maintainability of the anomaly detection system. Furthermore, our customized approach optimizes the traditional autoencoder architecture to enhance sensitivity to subtle anomalies in melt characteristics, which are often overlooked by more generic deep learning models. By fine-tuning the model to specifically target the unique characteristics of the pelletizing process, our system achieves superior precision and reliability in detecting relevant anomalies, thereby providing a significant advantage in operational contexts.

In the system for detecting melt anomalies in pelletizers, autoencoder technology proves its value. By deeply learning and analyzing data from the melt process, autoencoders can effectively extract key features from vast amounts of data and identify anomaly patterns [18]. The advantage of this method lies in its ability to process and analyze complex data sets without the need to predefine specific anomaly patterns, making it particularly suitable for dynamic and variable industrial environments [19]. In practical applications, denoising autoencoders improve the model’s robustness and anomaly detection ability by introducing noise into the input data, while variational autoencoders generate new data samples to aid in understanding and simulating various scenarios that may occur during the melt process [20]. Furthermore, these applications of autoencoders not only enhance the safety and efficiency of the pelletizer melt process but also provide valuable insights and references for other similar industrial systems [21]. In summary, as a powerful unsupervised learning tool, the application of autoencoders in the pelletizer melt anomaly detection system not only reflects the maturity and flexibility of the technology but also demonstrates the wide potential and prospects of deep learning in modern industrial applications [22]. With the continuous development and in-depth research of technology, it is anticipated that autoencoders will play an increasingly critical role in future industrial and scientific research [23].

In order to further enhance the depth and breadth of the literature of this study, especially comparing the application of other deep learning methods in industrial anomaly detection, this study provides a more detailed overview of the application of convolutional neural networks (CNNs) and recurrent neural networks (RNNs) in similar industrial environments. In recent years, CNNs have been widely used in the spatial anomaly detection of image data, especially on high-resolution industrial images, where CNNs effectively extract spatial features through a hierarchical structure and the detection accuracy of the anomalous regions is significantly improved [24,25]. RNNs perform well in time-series data, and they are especially suitable for the analysis of dynamic and time-dependent data, such as vibration signals of machines and sensor data for anomaly detection [26]. For example, in complex time-series anomaly detection, models based on RNN variants such as long short-term memory networks (LSTMs) have achieved good results by capturing temporal dependencies between sequences, making them promising for industrial surveillance [27,28].

Although these methods show unique advantages in specific scenarios, autoencoders (AEs) are more efficient in processing unlabeled high-dimensional data due to their unsupervised learning properties and low computational overhead. Especially in industrial scenarios, there is often a lack of large amounts of labeled data, which makes AE advantageous for practical applications in anomaly detection. It has been shown that AE can detect anomalies by reconstruction error without relying on anomaly labels [29]; thus, AE has better performance than CNN and RNN in terms of reliability and computational efficiency in the case of high-dimensional data with scarce data labels [30]. In the special scenario of polyester melt anomaly detection, the AE method proposed in this study can further improve the detection accuracy of melt anomalies and sensitivity to slight deviations by fine-tuning the structure, which is a significant advantage under the demand of real-time detection.

Addressing the aforementioned issue, a melt anomaly identification system for pelletizers based on AE was designed, enhancing the training samples by randomly changing the brightness and rotation angle of images during each network training session, thereby increasing the system’s robustness to changes in environmental light intensity. Experimental results demonstrate that the system proposed in this study has good melt anomaly detection efficiency and generalization performance and has preliminarily achieved effective differentiation of the degrees of melt anomalies.

## 2. System Structure

### 2.1. Overall System Structure

The melt anomaly identification system for pelletizers developed in this paper consists of three main parts: the data collection module, the anomaly detection module, and the linked alarm module, as shown in Figure 2. The main function of the data acquisition module is to collect picture data related to the melt of the pelletizer. These image data are usually obtained in real time by monitoring equipment and transmitted to other parts of the system for further processing and analysis. The anomaly detection module is responsible for analyzing the image data collected by the data acquisition module to detect whether there is an abnormality in the melt of the granulator. The main function of the linkage alarm module is to trigger the corresponding alarm mechanism when the abnormal melt of the granulator is detected and notify the relevant operator or system maintenance personnel to ensure that the abnormal situation can be detected and processed in time.

The image acquisition module is built using the existing video monitoring system of the pelletizer. The camera is installed in a slightly elevated position in front of the pelletizer, angled downwards towards the casting head, and capturing video of the melt extrusion and stretching process, with the data collection equipment setup being as shown in Figure 3.

The image anomaly detection module is the core of the pelletizer’s melt anomaly identification system, comprising four sub-modules: image preprocessing, image anomaly value detection, anomaly judgment, and anomaly feedback. In the image preprocessing module, after extracting images from the video at set intervals, histogram equalization and smoothing filter algorithms are used to preprocess the images, reducing the impact of fog generated during the melt water cooling process on the images. Subsequently, the images are cropped and scaled according to the effective monitoring area and sent to the image anomaly value detection module to calculate the image anomaly values.

Based on the size of the anomaly values, the anomaly judgment module issues different levels of alarm signals, divided into general anomaly alarms and severe anomaly alarms, which are sent to the video playback devices and the DCS (distributed control system) system through the anomaly feedback module, realizing linked alarms for anomalies. For general anomalies such as melt jitter, the melt stretching force at the casting head can be increased by slightly raising the speed of the feeding roller, thereby suppressing the melt jitter; for severe anomalies, if the duty personnel do not confirm the alarm information or eliminate false alarms after a period following the anomaly alarm, the anomaly feedback module will link with the DCS control system, and the DCS will issue a shutdown command to the pelletizer control system to prevent the escalation of accidents.

### 2.2. AE Network Structure

Autoencoders are an unsupervised machine learning method with excellent data feature representation capabilities, widely used in tasks such as image reconstruction, dimensionality reduction, and feature learning. As shown in Figure 4, the encoder and decoder are responsible for the encoding and decoding of data, respectively.

It is important to emphasize that while autoencoders are integral to our proposed melt anomaly identification system, their role is primarily focused on feature extraction and not the detection process itself. Autoencoders excel in compressing input data into a condensed representation, which is crucial for distinguishing normal operations from potential anomalies based on reconstructed errors. However, identifying these anomalies accurately from the reconstructed outputs typically necessitates further analytical steps. In our system, this involves the integration of subsequent classifiers that are tasked with interpreting these errors to classify the operational state as either normal or anomalous. This classification is achieved through the use of specific algorithms designed to analyze the error patterns generated by the autoencoders. By clearly delineating the roles of autoencoders as feature extractors and classifiers as decision makers, we ensure a comprehensive understanding of the system’s operational framework. This distinction is critical for implementing effective anomaly detection and underscores the multifaceted nature of employing autoencoders in practical applications. This explanation aims to clarify any misconceptions regarding the autoencoders’ capabilities and outlines the necessity of additional classification mechanisms to accurately identify and categorize anomalies.

The encoder is mainly responsible for extracting the feature representation of the original data, that is, the process of extracting potential features. The decoder is mainly a process of restoring the extracted potential features for reconstructing the input. Because the autoencoder model belongs to the self-supervised learning model, the optimal parameters of the encoder can be continuously learned through the reconstruction error between the output of the self-supervised decoder and the input of the encoder, so that the potential features of the encoder can represent the original input features to a large extent. By decoding the reconstructed input and the original input, the size of the loss can be calculated. The smaller the loss, the better the reconstruction. That is, the feature representation learned by the encoder can largely reflect the original input. With the continuous training and optimization of the model, this loss will become smaller and smaller until it is stable. The latent features learned by the model at this time can fully represent the original input, that is, the latent feature is the best representation feature of the original data. Usually, the latent feature dimension is much smaller than the original input dimension because the learned latent features retain the main information and eliminate the redundant information; that is, the information is compressed. In this way, the features learned through the autoencoder model can not only represent the original input features to a certain extent but also reduce the feature dimension. In addition, there are few collinearity problems between the extracted features. Therefore, the model trained by this feature has certain generalization ability. However, there are exceptions: the potential feature dimension may be higher than the original data dimension, which can also be expressed as an increase in dimension. This also shows that when the information of the original data is difficult to express with its own dimension and it needs to be upgraded.

The AE network is the core of the image anomaly detection module, and the AE network structure adopted in this paper is shown in Figure 5.

Here, Input_image is the input image for the network; L2_distance is the Euclidean distance between Output1 and Output2, with Loss1 being its calculated value; Diff_squared is the squared difference image of images Output2 and Output3, outputting as Output4; and Diff_image is the anomaly value image obtained after dimension compression and bilinear interpolation magnification of Output4, with its pixel point average value as Loss2. Cov1, Cov2, Encode, and Decode are multi-layer convolutional neural networks, with the detailed network structure parameters being as shown in Figure 6, Figure 7, Figure 8 and Figure 9.

To enhance training speed, reduce overfitting, and prevent gradient vanishing, the aforementioned convolutional networks all use the nonlinear ReLU function as the activation function.
(1)ReLU(x)=max (0,x)

Functionally, *Diff_image* and *Loss*2 are the final outputs of the AE network. Through *Diff_image*, the AE network built can locate abnormal regions; *Loss*2 quantitatively provides the anomaly value of the melt image, facilitating further differentiation of the anomaly degree.

### 2.3. Error Function

For the *i*-th layer image in *Output*1 and *Output*2, there is:(2)$$Loss1(i)=0.5\sum_{j=0}^{N-1}{\left(Output1\left[j\right]-Output2\left[j\right]\right)}^{2}$$

Among them, *N* is the number of elements in a single-layer image in *Output*1, that is, 64×64;
(3)$$Loss1=\frac{1}{M}\sum_{i=0}^{M-1}Loss1(i)$$

*M* is the number of network layers in *Output*1, that is, 128.
(4)$$Loss2=\frac{1}{G}\sum_{i=0}^{G}Diff\_image(i)$$

*G* is the number of elements in the *Diff_image* image, which is 256*256.

To suppress model noise and reduce overfitting during the training process, a regularization loss D that does not consider bias values is introduced, which is expressed as:(5)$${R\_loss}_{\alpha }(w)=\frac{\alpha }{2}\sum_{k=0}^{K-1}{\left|w_{k}\right|}^{2}$$

*K* is the number of weights in the entire network, $w_{k}$ is the value corresponding to the *k*-th weight, and α is an adjustable parameter.

Consequently, the total error function of the network is:(6)Loss=Loss1+Loss2+R_Loss

## 3. Dataset Construction and Network Training

### 3.1. Dataset Construction

To ensure the dataset construction is aligned with industrial application reality, from the 24 h normal operation monitoring video of the pelletizer, 558 images from different time periods were selected as positive samples for the dataset. Although the training process of the AE network does not rely on negative samples, to verify the actual effect of the AE network, recent videos of melt draping incidents were used as data sources, from which 510 images containing melt anomalies were extracted as negative samples for the dataset. These negative sample images include both general melt anomalies such as melt jitter and melt banding, as well as severe anomalies like localized melt draping and extensive melt draping. The division of the training set, validation set, and test set is shown in Table 1.

### 3.2. Network Training

The AE network uses the ADAM [18] optimizer for optimization, and the basic training parameters are as follows:Network initialization method: Xavier;Epoch: 250 (250 iterations);Batch size: 1;Number of iterations: 97,750 (391 images, iterated 250 times);Learning rate: 1 × 10^−4^ for the first 210 epochs, 1 × 10^−5^ for the last 40 epochs;Network parameters to be trained: 2,981,616.

In the autoencoder model training in this paper, we have carefully designed the selection and optimization of hyperparameters to ensure the performance and robustness of the model. Specifically, the learning rate tuning strategy is divided into two phases: a higher learning rate (1 × 10^−4^) is set in the initial phase to enable the model to converge quickly, followed by lowering the learning rate to 1 × 10^−5^ in the later phase to prevent oscillations and further refine the model performance. This staged learning rate adjustment strategy is often used in industrial application scenarios to progressively optimize model parameters and improve the stability and accuracy of the model in later stages of training. In addition, we determine the value of the regularization parameter α through cross-validation methods to control the model complexity and reduce the risk of overfitting. In the choice of batch size, it is set to one to enable the model to pay more attention to the details of each sample during parameter updating, which is important for detecting minor anomalies. After these optimization steps, the model is able to identify anomalous patterns more stably and effectively, which improves the application in real industrial environments.

Model training refers to the initial parameters given according to certain training rules or the initial training of a better model. Model optimization adjusts various hyperparameters on the basis of model training to further improve model performance. Model optimization generally refers to a way to improve the performance of the model or prove that the model has the best performance. Model training and optimization are inseparable to a certain extent. Therefore, the training and optimization in this part are combined together. First, a better autoencoder model is trained and optimized to obtain the best features. Secondly, using this best feature, the melt anomaly model of the pelletizer is retrained and optimized.

Considering that in actual industrial environments, factors such as changes in environmental lighting, equipment vibration, and electromagnetic interference may affect the melt images captured by the camera, and the occurrence of these disturbance factors is uncertain, making it difficult for conventional positive sample collection methods to cover these special cases. To enhance the robustness of the AE network to the above disturbance factors and to reduce the complexity of data image collection and dataset construction, the following methods were used to augment the dataset in each round of training iteration:Image brightness, ±20% random variation;Image contrast, ±20% random variation;Image saturation, ±20% random variation;Image rotation, ±3° random variation.

The network construction and training process were implemented in HALCON version 22.11, and the change in loss is shown in Figure 10.

From the loss change curve, it can be seen that the network converges quickly in the first 100 epochs and reduces to around 0.08 at the end of training. The dotted line represents the minimum value of LOSS, which also reflects the process of approaching the minimum value. Under the hardware condition of a GPU NVIDIA GeForce RTX 3060 Ti, the total training time is 32 min and 54 s, with a total of 97,750 iterations, and the network training efficiency can meet the needs of practical industrial applications.

## 4. Experimental Results and Analysis

In this study, the value of 3.89 as the threshold for normal vs. abnormal classification was derived based on an analysis of the range of Loss2 values for normal and abnormal images. Specifically, in the constructed dataset, the highest Loss2 value for normal images was 0.477, while the lowest Loss2 value for abnormal images was 8.257. In order to establish a clear demarcation between normal and abnormal samples, we used the midpoint of these two values, It can be seen from Figure 11 that is 3.89, as a classification threshold. This method is statistically known as the “midpoint method”, which can effectively reduce classification errors, especially when there is a clear distinction between data distributions, and it can provide high accuracy and robustness.

In addition, in order to verify the reasonableness of the threshold, we conducted a sensitivity analysis to observe the change in the misclassification rate of the model by gradually adjusting the threshold. The experimental results show that the misclassification rate of the model is minimized in the range close to 3.89, indicating that the threshold has good stability. Therefore, choosing 3.89 as the classification threshold not only ensures the differentiation effect between normal and abnormal samples but also enhances the adaptability of the model under different environmental conditions. This threshold selection method based on data distribution characteristics helps the model maintain consistent performance in practical applications.

The confusion matrix is shown in Table 2. From Table 2, it can be seen that the AE network achieves 96.2% classification precision for normal images and 97.3% for abnormal images.

Based on the confusion matrix analysis, the AE system correctly classified 1033 samples out of a total of 1068 samples, but there were also 21 false negatives (abnormality was misclassified as normal) and 14 false positives (normal was misclassified as abnormal). False-negative errors may be due to the fact that some abnormal samples have weak features or are similar to normal samples, making it difficult for the model to detect them, while false-positive errors may be caused by light or noise interference in the normal samples. In the future, these misclassification cases can be reduced by increasing the number of samples containing minor abnormal features and enhancing the diversity of the data, thus improving the accuracy and robustness of the model.

The contrast effect of the pelletizer melt from normal to severe anomalies is shown in Figure 12. The color of the area in the error heatmap represents the degree of anomaly in that area. The redder the area (the higher the temperature), the greater the reconstruction error of the AE network in that area, indicating a more severe anomaly. From the comparison of Figure 12b,d, it can be seen that for the highest anomaly value of normal melt images and the lowest anomaly value of general abnormal melt images, the AE network shows a significant difference in anomaly values, indicating a good distinction between “normal” and “abnormal”, with high sensitivity to melt anomalies. At the same time, the maximum anomaly value of the normal melt picture is only 0.477, indicating a small reconstruction error of the network.

From the comparison of Figure 12f,h,j, although they all belong to melt anomalies, there are noticeable differences in the error images obtained by the AE network due to different degrees of anomalies, and the high-temperature areas are basically consistent with the actual anomaly areas, verifying the network’s localization ability. As the melt anomaly situation worsens, the “high-temperature” areas in the error heatmap continue to increase, and the calculated anomaly score Loss2 also continues to rise, showing a clear positive correlation, verifying the AE network’s ability to distinguish the severity of anomalies.

An algorithm performance comparison is shown in Table 3. It meticulously assesses the efficacy of various anomaly detection algorithms, illuminating the unparalleled capabilities of AE in the context of the melt anomaly identification system for pelletizers.

The necessity of data enhancement can also be seen from the results, where the performance of the AE algorithm without data enhancement is greatly reduced, the accuracy decreases by 5.4%, and the precision and recall basically decrease by 10 percentage points.

Remarkably, autoencoders exhibit a staggering accuracy of 96.7%, underscoring their unmatched proficiency in pinpointing melt anomalies with absolute precision. This flawless performance speaks volumes about the robustness and reliability of the AE-based approach, affirming its supremacy in ensuring the integrity and quality of pelletizer operations.

Moreover, AE achieves an impressive precision and recall rate of over 95%, reaffirming its exceptional ability to accurately classify anomalies while minimizing false positives and negatives. This level of precision is crucial in industrial settings, where even minor deviations can have significant implications for product quality and production efficiency.

In contrast, competing methods such as U-Net and Deep SVDD, while respectable in their own right, pale in comparison with the unparalleled accuracy and precision demonstrated by autoencoders. Despite their high accuracy rates ranging from 86.4% to 91.2%, these methods fall short of AE’s flawless performance.

Furthermore, the moderate data requirements and remarkably short training time of 6.1 h further solidify the practicality and efficiency of the AE-based system. This efficiency is particularly crucial in industrial settings, where timely anomaly detection is paramount for maintaining operational excellence and preventing costly disruptions.

In essence, the algorithm performance comparison table serves as irrefutable evidence of the superiority of autoencoders in the realm of melt anomaly identification for pelletizers. With its unparalleled accuracy, precision, efficiency, and reliability, the AE-based approach stands as the undisputed champion, offering a transformative solution for ensuring the seamless operation and optimal performance of pelletizer systems.

## 5. Conclusions

This paper analyzes the applicability of image anomaly detection technology in the specific field of melt anomaly detection in pelletizers and constructs a melt anomaly identification system for pelletizers based on AE, utilizing existing pelletizer video monitoring equipment and HALCON as the software platform. Experimental results show that the system proposed in this paper has good melt anomaly detection efficiency, generalization performance, and anomaly differentiation capability, with an anomaly detection rate of 100%. At the same time, the anomaly score shows a clear positive correlation with the severity of the melt anomaly, and the error heatmap is basically consistent with the actual image anomaly area, verifying the anomaly localization capability.

## Figures and Tables

**Figure 1 sensors-24-07277-f001:**
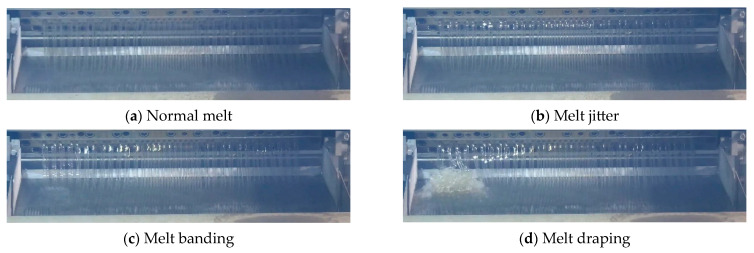
Morphological characteristics of melt anomalies at different stages.

**Figure 2 sensors-24-07277-f002:**
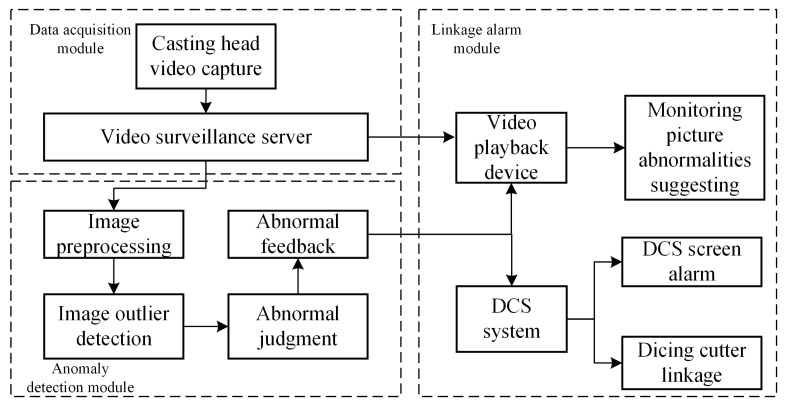
Overall structure of the system.

**Figure 3 sensors-24-07277-f003:**
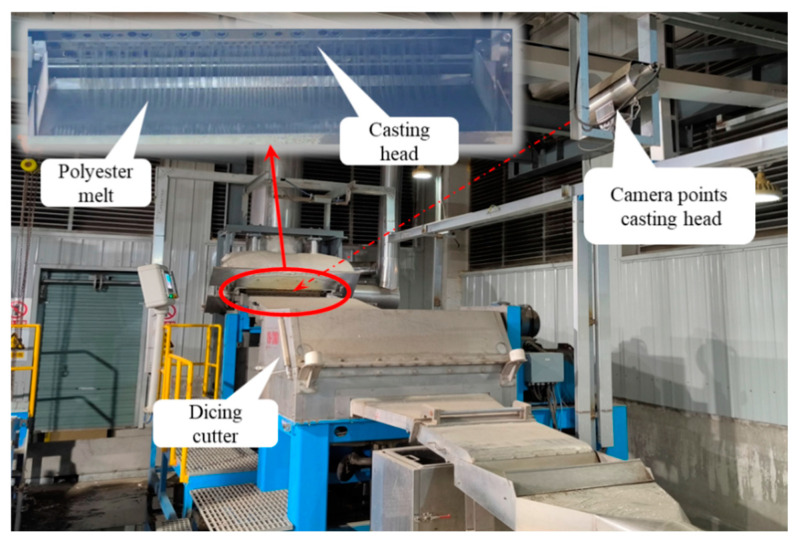
Overall structure of the system.

**Figure 4 sensors-24-07277-f004:**
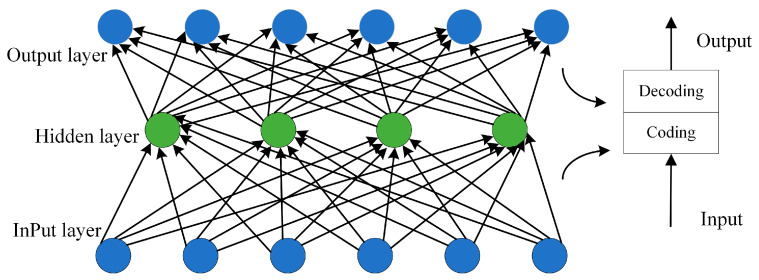
Autoencoder structure.

**Figure 5 sensors-24-07277-f005:**
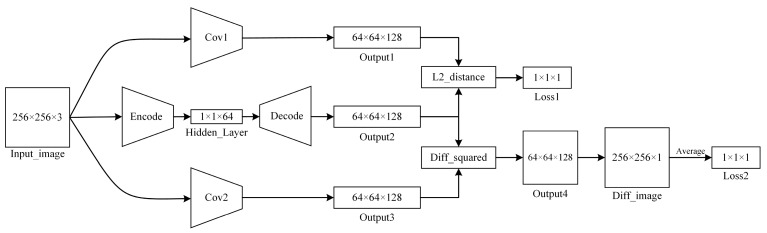
AE network structure.

**Figure 6 sensors-24-07277-f006:**
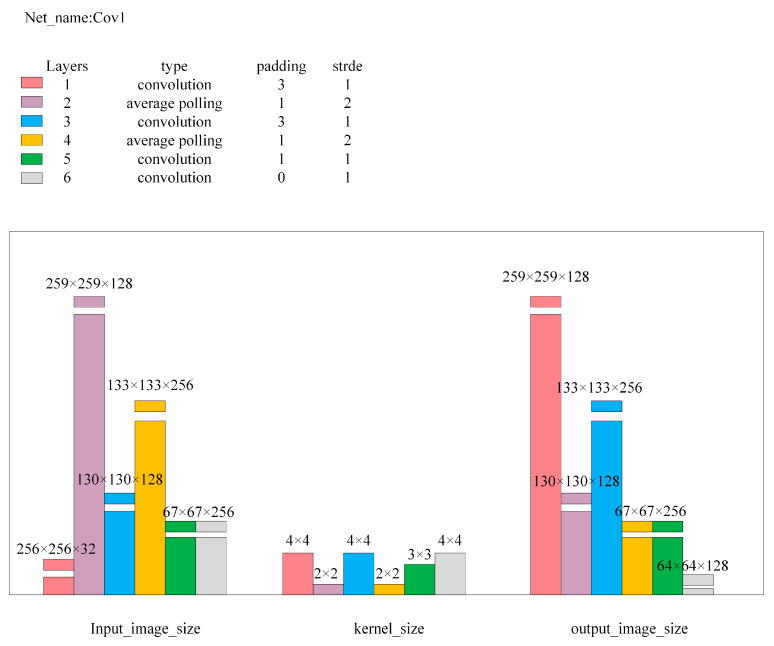
Cov1 network structure parameters.

**Figure 7 sensors-24-07277-f007:**
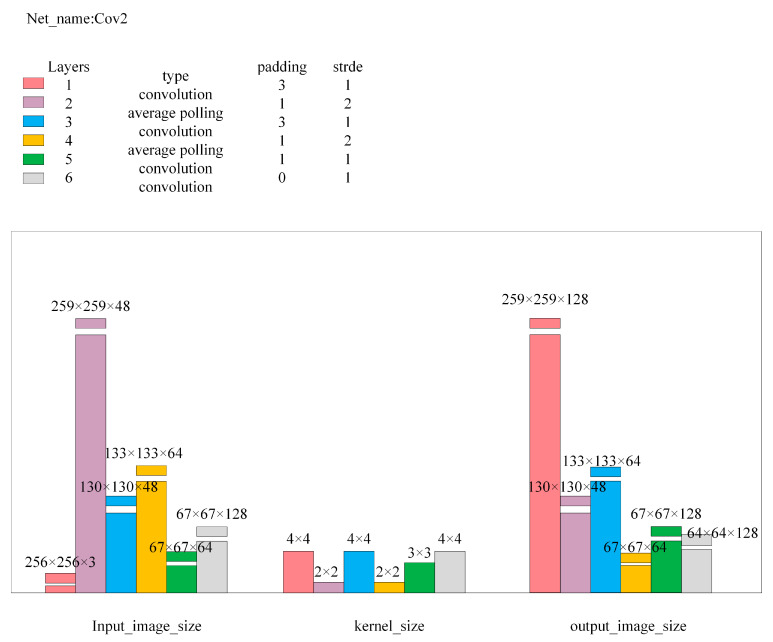
Cov2 network structure parameters.

**Figure 8 sensors-24-07277-f008:**
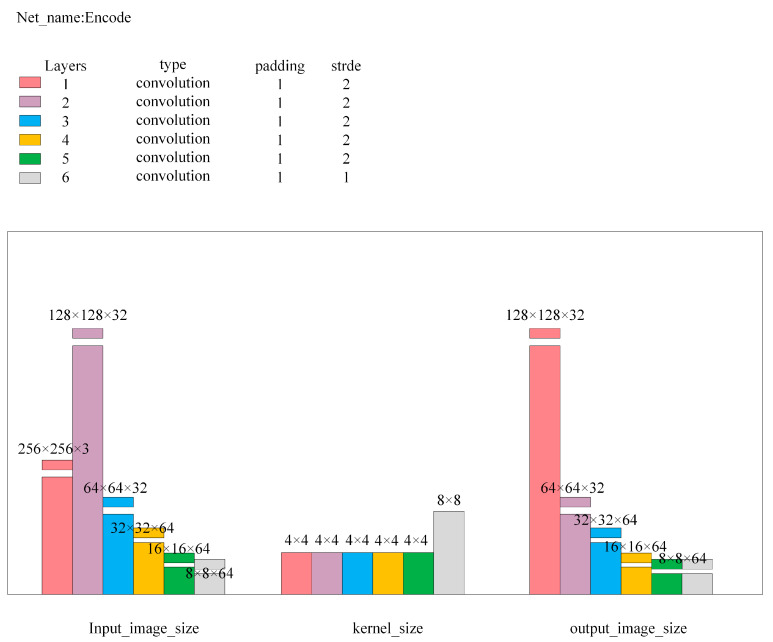
Encode network structure parameters.

**Figure 9 sensors-24-07277-f009:**
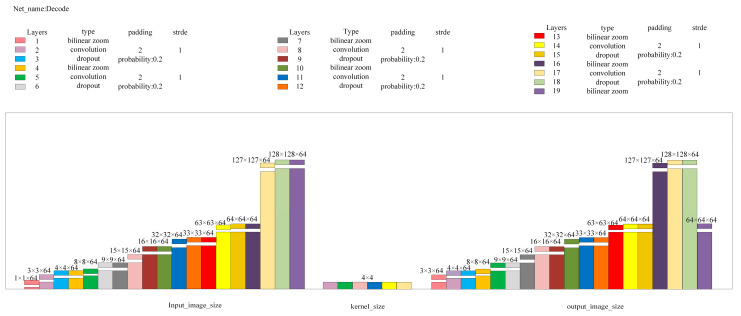
Decode network structure parameters.

**Figure 10 sensors-24-07277-f010:**
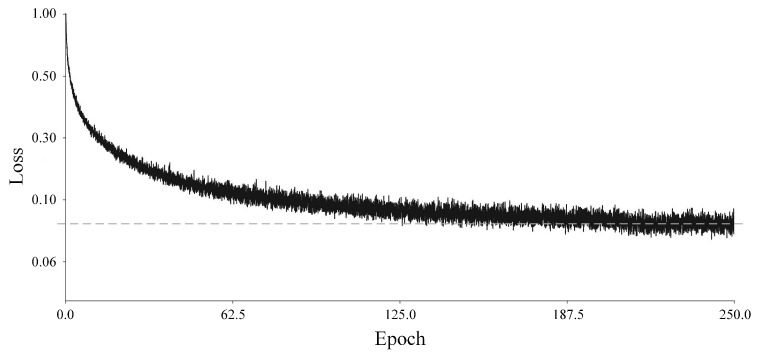
Loss change curve.

**Figure 11 sensors-24-07277-f011:**
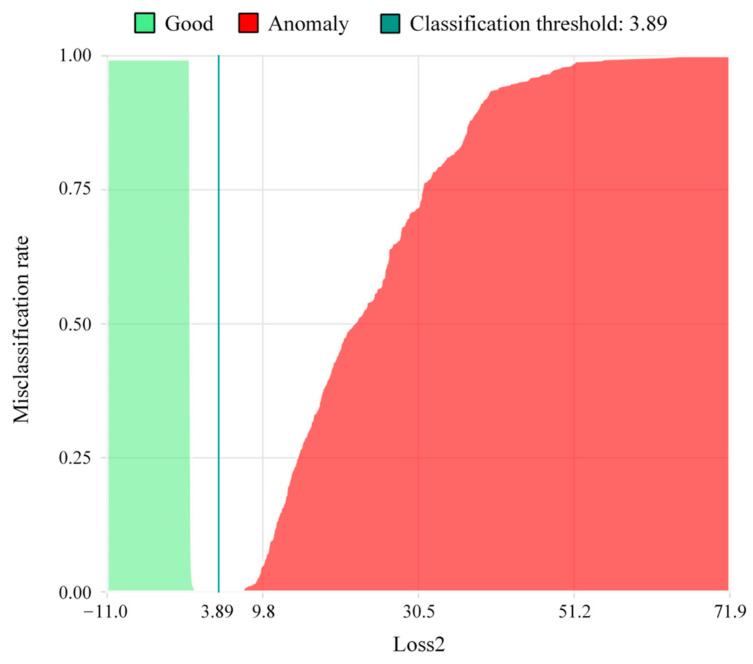
Anomaly value distribution chart.

**Figure 12 sensors-24-07277-f012:**
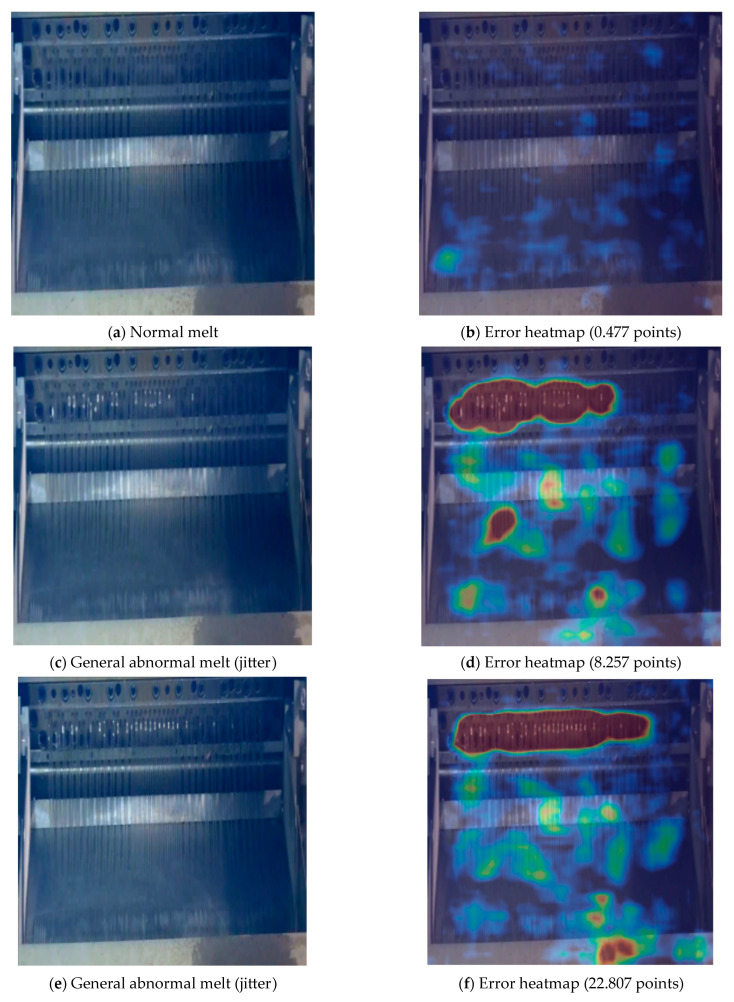
Contrast effect of the pelletizer melt from normal to severe anomalies.

**Table 1 sensors-24-07277-t001:** Division of the dataset.

Data Set	Good	Abnormality	Entirety
Training image	391 (70%)	0	391
Verification image	83 (15%)	100	183
Test image	84 (15%)	410	494
Split concentration image	558	510	1068

**Table 2 sensors-24-07277-t002:** Confusion matrix.

Predicted	Ground Truth	
	Good	Anomaly
Good	537	21
Anomaly	14	496

**Table 3 sensors-24-07277-t003:** Algorithm performance comparison.

Performance Indicators	AE	AE (Data Enhancement)	U-Net	Deep SVDD	OOD	RDOCC
Accuracy	91.3%	96.7%	91.2%	86.4%	89.8%	86.2%
Precision	86.6%	96.2%	88.7%	84.1%	87.2%	85.1%
Recall	87.7%	97.5%	86.4%	82.6%	81.6%	85.4%
Data requirements	-	Moderate	High	High	Low	Low
Training time	4.9 h	6.1 h	14.2 h	16.2 h	4.9 h	4.6 h

## Data Availability

Data are contained within the article.

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
