# Peer review of "Autoencoder-Based System for Detecting Anomalies in Pelletizer Melt Processes"

_sensors, 2024, doi:10.3390/s24227277_

Round 1

Reviewer 1 Report

Comments and Suggestions for Authors

This paper addresses the problem of anomaly detection in industrial melt processes and proposes an autoencoder-based detection system.

The paper has several strengths:

(1) Practical Value

The anomaly detection in the polyester pelletizing melt process, a critical step in industrial production, holds significant practical importance. By addressing the limitations of manual monitoring and traditional image processing methods, the study contributes to improving production efficiency and product quality.

(2) Feasibility of the Technical Approach

Autoencoders are widely used in unsupervised learning, particularly suitable for industrial scenarios with scarce labeled data. Based on this, the paper designs a system capable of real-time anomaly detection, theoretically solving the common issue of label scarcity in industrial environments.

(3) Comprehensive Experiments

Through a series of experiments, the authors validate the model's effectiveness and provide detailed comparisons with other deep learning models, demonstrating the advantages of autoencoders in terms of accuracy, recall, and training time.

However, the paper has some shortcomings:

(1) Insufficient Analysis of Related Research

Although some relevant literature is cited, the depth and breadth of references in the background and methodology sections need improvement. For instance, the application of other deep learning methods (e.g., CNNs or RNNs) in industrial anomaly detection could be further explored to better highlight the unique advantages of the autoencoder approach.

(2) Unclear Theoretical Methodology

The paper lacks detailed explanations regarding hyperparameter selection and model optimization, such as the learning rate adjustment strategy and the basis for selecting the regularization parameter α. Additionally, while the authors applied data augmentation techniques (e.g., variations in brightness, contrast, saturation, and rotation), the specific impact of these augmentations on model performance is not discussed. It is recommended to provide a comparative analysis of model performance before and after data augmentation to demonstrate its necessity and effectiveness.

(3) Insufficient Analysis of Experimental Results

The fixed anomaly detection threshold (e.g., 3.89) lacks an explanation of its selection basis. A sensitivity analysis on threshold selection would ensure the model's robustness in different environments. Furthermore, the model's effectiveness is validated primarily in a single scenario, lacking cross-scenario or cross-device generalization tests. This is crucial in industrial applications, and it is suggested to include additional experiments or discussions on this aspect.

(4) Lack of Model Innovation

Autoencoders have been widely applied in unsupervised anomaly detection. While this paper makes a valuable contribution by applying the method in a specific industrial context, it does not introduce any optimizations or improvements to the standard autoencoder architecture. The methodological innovation is thus limited.

Comments on the Quality of English Language

The quality of English in this paper is acceptable but could be improved in several areas to enhance clarity and readability. Below are specific comments and examples:

(1)Grammar and Syntax

Some grammatical issues affect the flow of the paper. For instance:

In the Abstract, the sentence "Experimental results indicate that the system proposed has good melt anomaly detection efficiency and generalization performance, and has effectively differentiated degrees of melt anomalies." could be revised to:

"Experimental results indicate that the proposed system demonstrates good efficiency in melt anomaly detection and generalization, effectively differentiating between degrees of melt anomalies."

(2)Word Choice and Tone

In the Introduction, the phrase "staggering accuracy" (page 15, line 370) is overly emphatic and could be replaced with "high accuracy" to maintain a more formal tone.

(3)Sentence Structure

Some sentences are overly long and could be broken into shorter sentences for better readability. For example:

On page 3, lines 85–91:

"Compared to deep learning-based anomaly detection algorithms like Deep SVDD, OOD, and RDOCC, the AE-based anomaly detection algorithm relies only on normal samples during training, avoiding the difficulty of extensively collecting anomaly samples."

This could be rewritten as:

"The AE-based anomaly detection algorithm relies solely on normal samples during training. This avoids the challenge of collecting a large number of anomaly samples, which is required by other deep learning-based algorithms such as Deep SVDD, OOD, and RDOCC."

(4)Consistency

Terminology such as "melt anomalies" and "anomaly detection" should be consistently used throughout the paper. On page 11, line 294, the phrase "split concentration image" is unclear and inconsistent with standard terminology. Consider revising for clarity.

(5)Formal Academic Style

Some sections could adopt a more formal tone. For example, in the Results section, phrases such as "unparalleled accuracy" (page 15, line 369) should be toned down to maintain objectivity. A better alternative could be "outstanding accuracy compared to other methods."

Reviewer 2 Report

Comments and Suggestions for Authors

The paper presents a novel approach for detecting anomalies in the melt process of polyester pelletizers using an autoencoder-based system. It highlights the challenges of traditional monitoring methods, including manual detection, and proposes an AE-based image anomaly detection system to improve efficiency and accuracy. The authors implemented the model with additional mechanisms for robustness against environmental variations, enhancing generalization and accuracy. The experimental results indicate 100% accuracy in anomaly detection, significantly improving over traditional methods.

Strengths:

1. The paper applies autoencoder technology, typically used for anomaly detection, in an industrial context where manual monitoring is prevalent.

2. The data augmentation strategies, including random brightness and rotation adjustments, contribute to the model's robustness.

Drawbacks:

1. Although the AE system performs well, the study does not adequately compare it with other state-of-the-art methods beyond a few baseline algorithms. Models like CNNs and RNNs are mentioned in theory but not implemented for direct comparison, which could enhance the robustness of the claims.

2. The dataset size is relatively small (558 images for normal operations and 510 images for anomalies). This small sample size may not generalize well, especially for complex industrial applications with high variability.

3. The paper lacks a discussion of potential limitations of the AE-based approach, such as its susceptibility to overfitting due to small datasets or difficulty distinguishing subtle anomalies in high-noise environments.

4. The paper does not discuss any error analysis or cases where the model may have been misclassified, which could provide insights into the model's limitations and potential improvements.

Recommendations:

1. Implementing and comparing CNN- and RNN-based models directly against the AE system would strengthen the claims regarding AE's superiority and computational efficiency.

2. Incorporate a section analyzing cases where the AE system may misclassify images. This would provide insights into the types of anomalies that the model struggles with and could guide future improvements.
